# Optical gap and fundamental gap of oligoynes and carbyne

Johannes Zirzlmeier[1,9], Stephen Schrettl [2,6,9], Jan C. Brauer[2], Emmanuel Contal [2,7], Laurent Vannay[3], Éric Brémond [3,8], Eike Jahnke[4], Dirk M. Guldi [1✉], Clémence Corminboeuf [3✉], Rik R. Tykwinski [5✉] & Holger Frauenrath [2✉]

The optoelectronic properties of various carbon allotropes and nanomaterials have been well established, while the purely *sp*-hybridized carbyne remains synthetically inaccessible. Its properties have therefore frequently been extrapolated from those of defined oligomers. Most analyses have, however, focused on the main optical transitions in UV-Vis spectroscopy, neglecting the frequently observed weaker optical bands at significantly lower energies. Here, we report a systematic photophysical analysis as well as computations on two homologous series of oligoynes that allow us to elucidate the nature of these weaker transitions and the intrinsic photophysical properties of oligoynes. Based on these results, we reassess the estimates for both the optical and fundamental gap of carbyne to below 1.6 eV, significantly lower than previously suggested by experimental studies of oligoynes.

[1] Department of Chemistry and Pharmacy & Interdisciplinary Center for Molecular Materials (ICMM), Friedrich-Alexander-Universität Erlangen-Nürnberg (FAU), Egerlandstraße 3, 91058 Erlangen, Germany. [2] Ecole Polytechnique Fédérale de Lausanne (EPFL), Institute of Materials, Laboratory of Macromolecular and Organic Materials, EPFL – STI – IMX – LMOM, MXG 037, Station 12, 1015 Lausanne, Switzerland. [3] Ecole Polytechnique Fédérale de Lausanne (EPFL), Institute of Chemical Science and Engineering, Computational Molecular Design Laboratory EPFL – SB – ISIC – LCMD, BCH 5312, 1015 Lausanne, Switzerland. [4] Department of Chemistry and Pharmacy & Interdisciplinary Center for Molecular Materials (ICMM), Friedrich-Alexander-Universität Erlangen-Nürnberg (FAU), Nikolaus-Fiebiger-Straße 10, 91058 Erlangen, Germany. [5] Department of Chemistry, University of Alberta, Edmonton, AB T6G 2G2, Canada. [6] Present address: Adolphe Merkle Institute, University of Fribourg, Chemin des Verdiers 4, 1700 Fribourg, Switzerland. [7] Present address: Institute UTINAM, UMR CNRS 6213, University of Bourgogne Franche-Comté, 16 Route de Gray, 25030 Besançon, France. [8] Present address: Université de Paris, ITODYS, CNRS, F-75006 Paris, France. [9] These authors contributed equally: Johannes Zirzlmeier, Stephen Schrettl. ✉email: dirk.guldi@fau.de; clemence.corminboeuf@epfl.ch; rik.tykwinski@ualberta.ca; holger.frauenrath@epfl.ch

Organic molecules and polymers with π-conjugated systems have properties that are remarkably different from typical saturated organic compounds, such as extensive electron delocalization, a small HOMO-LUMO gap, and large polarizability[1–3], which render them useful for optoelectronic applications[4]. This has been particularly showcased for carbon-rich molecules and nanostructures with extended $sp^2$-hybridized carbon frameworks, such as fullerenes, carbon nanotubes, or graphene[5]. Carbon-rich structures comprising chains of more than a few $sp$-hybridized carbon atoms, on the other hand, are thermodynamically unstable and remain challenging synthetic targets[6]. Although this inherent reactivity toward rearrangement reactions to other carbon allotropes has inspired the development of approaches for the preparation of carbon nanomaterials that proceed under very mild conditions[7–10], it has rendered the $sp$-hybridized carbon allotrope "carbyne" $(C \equiv C)_\infty$ effectively inaccessible[11].

The unabated interest in the fundamental properties of this missing carbon allotrope, however, has stimulated significant research efforts following the oligomer approach[1,12,13], which aims at estimating the physical properties from the saturation values or extrapolated asymptotic limits of homologous series of oligoynes R–$(C \equiv C)_n$–R referred to subsequently as **R[$n$]**[14–19]. Various series of stable oligoynes have been prepared by using bulky end groups or by encapsulation in the form of rotaxanes[15–17,20–22]. Their highest wavelength absorption maxima observed in the ultraviolet–visible (UV-Vis) spectra of solutions have often been used in these studies, namely to extrapolate the optoelectronic properties of carbyne. Depending on the model used for such extrapolations, estimated values of 2.18–2.56 eV have been reported for the absorption onset of carbyne[15–19]. The vast majority of investigations have, however, focused on analyzing the most intense optical bands that dominate the spectra of oligoynes ($\lambda_{main}$). On the other hand, weak bands ($\lambda_{weak}$) are observed at higher wavelengths (lower energy) than the $\lambda_{main}$ bands in solution measurements of a range of oligoyne derivatives with various end groups[23–28], most commonly described empirically without concern for their origin. In cases in which $\lambda_{weak}$ bands are directly addressed, their origin has been dismissed as the result of impurities[26], or attributed to orbital mixing with the terminal substituents[16,19]. Two systems have been examined in detail, the parent series of oligoynes **H[$n$]** and diaryl oligoynes. Characteristics of the **H[$n$]** series, whereas foundational to the present study, are unique due to the lack of terminal substituents, which results in $D_{\infty h}$ symmetry for all members of the series irrespective of length[29–31]. Moreover, analyses of the latter are complicated by significant conjugation to, and molecular symmetry considerations resulting from, the aromatic end groups[32–34]. Finally, the $\lambda_{weak}$ bands have been detailed for a singular example of an alkyl-endcapped hexayne[35]. Computational analyses of electronic absorption spectra of representative oligoynes suggest that these bands can be attributed to two overlapping forbidden electronic transitions of the oligoyne moieties[31–41]. In total, however, previous studies have either been limited to a single compound or hampered by derivatives whose properties were dominated or perturbed by the presence of terminal substituents.

With the present work, we address the optoelectronic properties of oligoynes and carbyne in a way that clarifies the issues encountered in previous works. We report steady-state and time-resolved spectroscopic analyses, as well as computations, on two homologous series of oligoynes with non-conjugated end groups (Fig. 1a). Our results show that the weak intensity absorptions at higher wavelengths, $\lambda_{weak}$, are indeed intrinsic photophysical features of oligoynes that can be assigned to a weakly allowed transition ($S_0 \rightarrow S_{2/3}$), whereas the main optical absorptions $\lambda_{main}$

correspond to an even higher transition ($S_0 \rightarrow S_n$), and the lowest-energy $S_0 \rightarrow S_1$ transition remains completely dipole-forbidden. We find, thus, that the optical gap of oligoynes ($E_{opt}$) and, by extension, of carbyne is on the order of 1.5–1.6 eV, which is significantly smaller than reported in all previous studies and, moreover, provides an upper boundary for the fundamental gap of carbyne that is lower than that of $sp^2$-hybridized poly(acetylene)[42–44].

## Results and discussion

**Optical versus fundamental gap and symmetry considerations.** As recently emphasized by Bredas[45], and particularly appreciated in solid state physics[46,47], the notion of a "band gap" has often been used broadly to correlate experimental data obtained by different techniques, such as photoemission, photoconduction, or optical absorption, with the different underlying physical processes. One should, however, distinguish between the "optical gap", $E_{opt}$, and the "fundamental gap", $E_g$ (Fig. 1b, c). The optical gap is the lowest-energy transition observed in the experimental absorption spectra corresponding to the vertical excitation energy from the ground state to the first dipole-allowed excited state. By contrast, the fundamental gap is defined as the difference between the energies of the first ionization potential and the first electron affinity ($EA$)[45,48].

A comparative discussion of the experimentally accessible values of $E_{opt}$ and $E_g$ on the basis of excitonic states must consider molecular symmetry arguments. For the hypothetical carbon allotrope carbyne $(C \equiv C)_\infty$ with a sufficiently large number of carbon–carbon triple bonds, the derivatives with $(C \equiv C)_n$ and $(C \equiv C)_{n+1}$ are indistinguishable in their properties. The corresponding point group, in the ideal (linear) conformation, is expected to be $D_{\infty h}$ with degenerate frontier molecular orbitals that possess $\Pi_u$ and $\Pi_g$ symmetries (Fig. 1b). A transition between these frontier orbitals will lead to a splitting of the excitation energy levels into three distinct electronic states: one degenerate state, $\Delta_u$, as well as two non-degenerate states, $\Sigma^+_u$ and $\Sigma^-_u$. The dipole-allowed representations ($\mu_{xyz}$) as given by the $D_{\infty h}$ character table are $\Pi_u$ and $\Sigma^+_u$. Accordingly, the symmetry-allowed transition of lowest energy should populate the lowest-energy, dipole-allowed $\Sigma^+_u$ state, which should therefore be defined as the optical gap $E_{opt}$ of carbyne (Fig. 1b). The lowest-lying, but dipole-forbidden transition to the $\Sigma^-_u$ state (which is hence undetectable in UV-Vis spectra) must be lower in energy than the fundamental gap $E_g$. The difference is the exciton binding energy, which is substantial in organic molecules and materials (Fig. 1b).

The presence of terminal substituents R in oligoynes R–$(C \equiv C)_n$–R (**R[$n$]**) perturbs the energy profile when compared with the parent compounds H–$(C \equiv C)_n$–H (**H[$n$]**) and therefore alters the symmetry rules. Computational analyses of oligoynes endcapped with two identical substituents $CR_3$ show the point group $D_3$ (owing to free rotation of the end groups), with frontier orbitals of $E$ symmetry (Fig. 1c). Again, splitting of the excitation energy levels upon an electronic transition yields three electronic states, one degenerate state with $E$ symmetry and two non-degenerate states $A_1$ and $A_2$, of which only $A_2$ and $E$ are dipole-allowed representations ($\mu_{xyz}$) according to the $D_3$ character table. Qualitatively similar to the situation in carbyne, the lowest-energy transition to the $A_1$ state remains symmetry-forbidden while transitions to the non-degenerate $A_2$ state are symmetry-allowed. Different from carbyne, however, the transition to the degenerate state $E$ now becomes (weakly) allowed owing to the lower symmetry of the molecules ($D_3$ versus $D_{\infty h}$). This should result in a series of weak absorption bands in the experimental

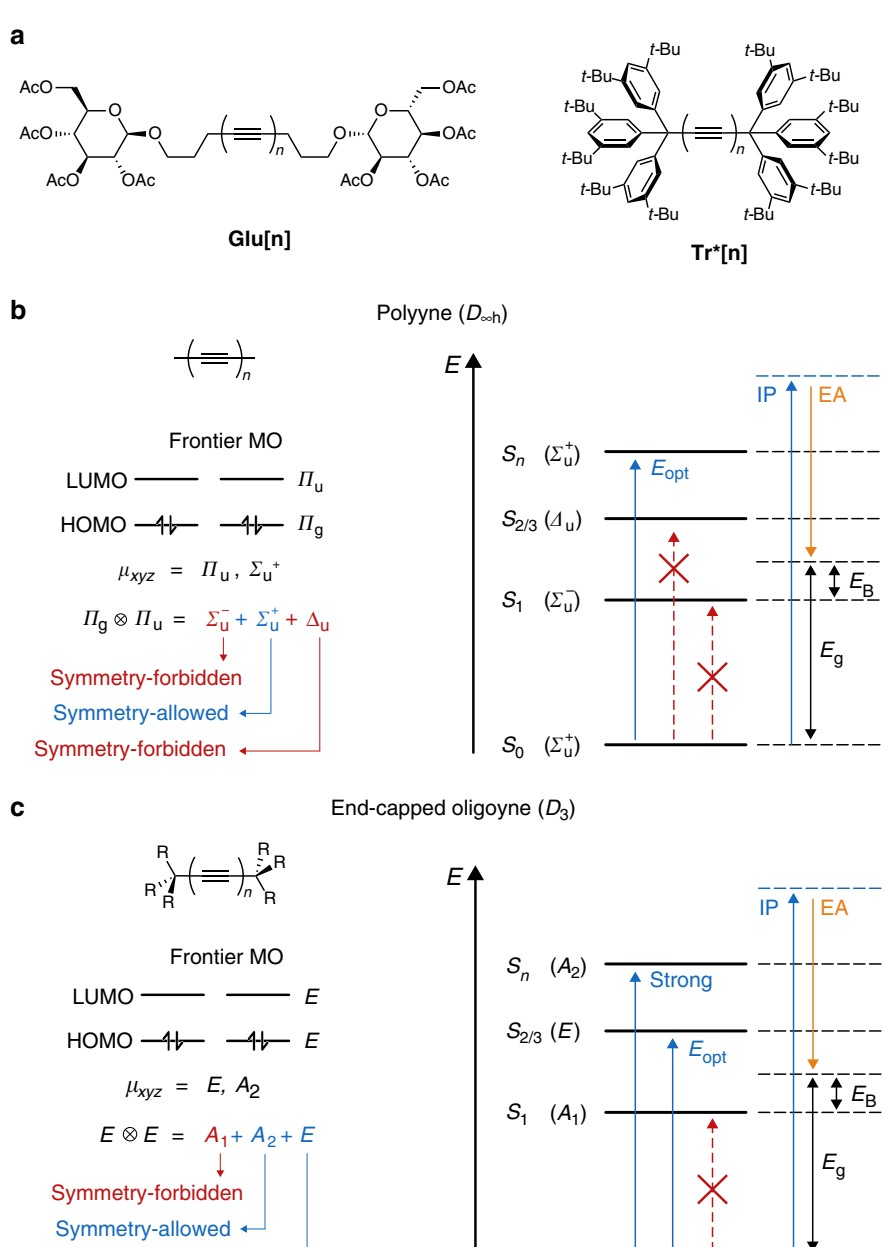

**Fig. 1 Oligoynes and their optoelectronic properties. a** Chemical structures of the $C_2$-symmetric glycosylated oligoynes **Glu[n]**, as well as the $D_3$-symmetric oligoynes **Tr*[n]** featuring triarylmethyl endcapping groups. **b, c** Graphical representations of the symmetry arguments for the electronic transitions and their respective consequences for the optical gap $E_{opt}$ and the fundamental gap $E_g$ for **b** polyyne ($D_{\infty h}$) as well as **c** typical oligoynes $R_3C-(C\equiv C)_n-CR_3$ (with the ionization potential $IP$, the electron affinity $EA$, the $S_0 \rightarrow S_n$ transition (strong), and the electron-hole pair binding energy $E_B$)[45].

UV-Vis spectra of oligoynes at lower energy and accordingly lead to a drastically altered optical gap $E_{opt}$ than that estimated from the $A_2$ transitions (Fig. 1c). When the end groups impose an even lower symmetry (e.g., $C_2$ point group), the degeneracy of the frontier orbitals is marginally lifted, resulting in one main (bright) optical transition in experimental spectra along with several further weak transitions at lower energies (see Wakabayashi et al.[31] for a detailed analysis of the relevant point groups). In any case, these bands are expected to disappear at extended chain lengths as the influence of the terminal substituents is decreased, but even minor deviations from an idealized linear conformation will render them weakly allowed[27].

These considerations motivated us to perform a detailed investigation of the solution-phase spectroscopic properties of oligoyne derivatives by experimental and computational means (a detailed description of the methodologies is given in the Methods section). We chose two series of oligoynes with terminal groups of different chemical nature and symmetry, namely the $C_2$-symmetric glycosylated oligoynes **Glu[n]** and the $D_3$-symmetric oligoynes **Tr*[n]** featuring triarylmethyl end groups (Fig. 1a)[17,22]. The terminal groups are not electronically conjugated with the oligoyne segments in either series, and the observed photophysical properties arise from the oligoynes and not from end-group effects.

**Steady-state UV-Vis spectroscopy.** The UV-Vis absorptions of the oligoynes **Glu[n]** (in acetonitrile or dichloromethane solutions) and **Tr*[n]** (in a hexane solution) show a red-shift with increasing number of triple bonds $n$ and display

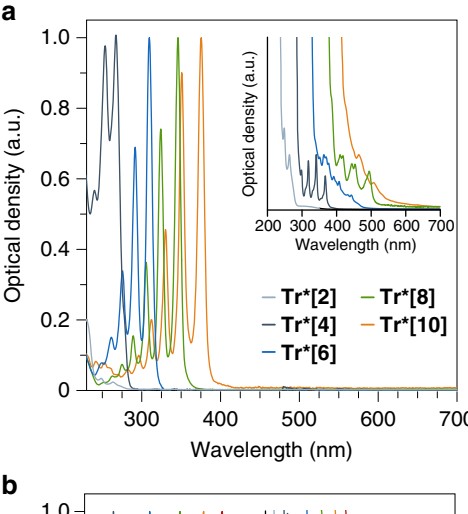

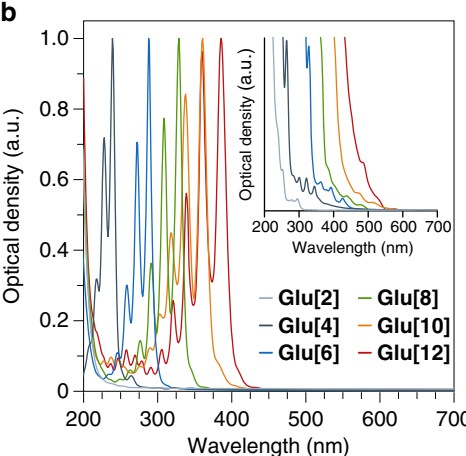

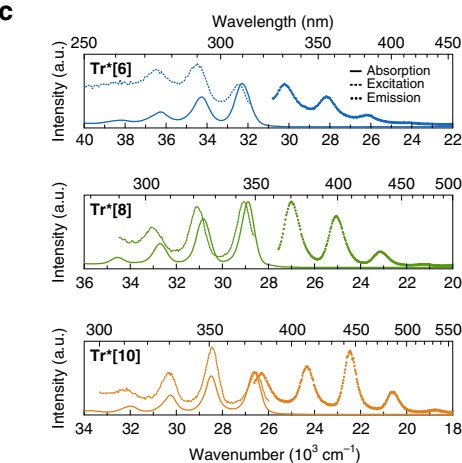

**Fig. 2 Absorption and emission spectra of oligoynes.** Normalized absorption spectra of **a** the **Tr\*[n]** and **b** the **Glu[n]** series recorded in hexane and acetonitrile, respectively. The insets show the additional bands with weak intensity at wavelengths higher than the longest wavelength absorption maximum ($\lambda_{main}$). **c** Steady-state absorption (solid line), emission (em., dotted line), and excitation (exc., dashed line) spectra in hexane of **Tr\*[6]** (em. $\lambda_{exc}$ = 290 nm; exc. $\lambda$ = 355 nm), **Tr\*[8]** (em. $\lambda_{exc}$ = 320 nm; exc. $\lambda$ = 400 nm), and **Tr\*[10]** (em. $\lambda_{exc}$ = 330 nm; exc. $\lambda$ = 446 nm). Source data are provided as a Source Data file.

a characteristic vibronic fine structure starting approximately with **Glu[4]** and **Tr\*[4]** (Fig. 2a, b; Supplementary Fig. 1). The spectra of the longer oligoynes display several well-defined maxima, among which the highest wavelength absorption,

$\lambda_{main}$, exhibits the highest intensity (Table 1; Supplementary Table 1).

For example, $\lambda_{main}$ for **Glu[n]** shifts from 240 nm for **Glu[4]** to 390 nm for **Glu[12]**. At the same time, the vibronic progressions ($\Delta E$ ($S_0 \rightarrow S_n$) in Table 1) decrease from ~2000 cm$^{-1}$ for **Glu[4]** to ~1800 cm$^{-1}$ for **Glu[12]**, reflecting the known trend of greater conjugation as oligoyne length is increased[49]. Moreover, the molar absorption coefficient, $\varepsilon_{main}$, increases significantly with the number of C≡C bonds (Supplementary Table 1). A comparison of the two oligoynes series indicates that $\lambda_{main}$ of the **Tr\*[n]** series is slightly red-shifted by 10–30 nm compared with that of **Glu[n]**. The difference in absorption energies between the two series diminishes as a function of length[13], and can likely be attributed to a combination of solvatochromic effects and to hyperconjugation, i.e., the orbital overlap between the conjugated π-system and the end groups. The latter is well documented in the literature[49], and particularly observed in the computational analysis of the orbitals participating in the main optical transitions of the **Tr\*[n]** series of oligoynes (vide infra). All of these observations are in excellent agreement with previous reports on other series of oligoynes[13–18]. A closer inspection of the absorption spectra of **Glu[n]** and **Tr\*[n]** reveals additional absorption bands with weak intensities ($\lambda_{weak}$), at wavelengths significantly higher than $\lambda_{main}$ (Fig. 2a, b, insets). The highest wavelength absorption maximum of this series of bands is red-shifted by >100 nm with respect to the main absorption maxima (Table 1). No significant changes are observed in temperature-dependent UV-Vis spectra, so that aggregation of the oligoynes in solution can be excluded as the origin of $\lambda_{weak}$ (Supplementary Fig. 2).

As we will elucidate in the following sections, we infer that the $S_1$ state remains dark owing to the dipole-forbidden character of the $S_0 \rightarrow S_1$ transition, in agreement with previous computational analyses and experimental studies of oligoynes[30,31,35]. We hence attribute the series of "weak signals" to the spectroscopically weakly allowed $S_0 \rightarrow S_{2/3}$ transition to the degenerate state $E$. In accordance with this interpretation, the amplitudes of $\lambda_{weak}$ absorptions are two to three orders of magnitude lower relative to the main optical absorptions, $\lambda_{main}$, that can hence tentatively be assigned to a higher $S_0 \rightarrow S_n$ transition.

**Fluorescence spectroscopy.** Although the fluorescence of oligoynes is generally weak[38], we successfully recorded fluorescence and excitation spectra of the **Tr\*[n]** series in hexane solutions (Fig. 2c). The fluorescence spectra are almost mirror images of the $\lambda_{main}$ bands in the absorption spectra, and the corresponding excitation spectra are exact matches of the latter. Considering that the weakly allowed and the dipole-forbidden transitions are non-emissive and, in any case, too low in energy to contribute to the emission bands, the fluorescence originates only from the tentatively assigned $S_n \rightarrow S_0$ transition. Notably, these oligoynes are thus apparently among the few examples where fluorescence does not obey Kasha's rule, that is, emission occurring from the vibrationally relaxed lowest excited state[50]. The large relevant energy gaps (e.g., ~8000 cm$^{-1}$ or 1 eV for **Tr\*[8]**) and low oscillator strengths of the fundamental transitions render either thermal repopulation or reverse internal conversion processes unlikely, so that prompt fluorescence from the excited state remains as the presumed dominant deactivation pathway of photoexcited oligoynes[51].

From the $S_0 \rightarrow S_n$ absorptions and shortest wavelength fluorescence, Stokes shifts as small as 280 cm$^{-1}$ are derived (Fig. 2c, Supplementary Fig. 3), which can be explained with the structural rigidity of oligoynes and little bond length changes upon excitation. Although small Stokes shifts are typically correlated with high fluorescence quantum yields, $\Phi_F$, we observe

**Table 1 Experimental data determined by UV-Vis absorption spectroscopy and by global analysis of the transient absorption spectra.**

| Compound | UV-Vis spectroscopy | | | Transient absorption spectroscopy | | | |
|---|---|---|---|---|---|---|---|
| | $S_0 \rightarrow S_n$ ($\lambda_{main}$) [nm] | $\Delta E$ ($S_0 \rightarrow S_n$) [cm$^{-1}$] | $S_0 \rightarrow S_{2/3}$ ($\lambda_{weak}$) [nm] | $\tau_1$ [ps] | $\tau_2$ [ps] | $\tau_3$ (Ar) [µs] | $\tau_3$ (O$_2$) [ns] |
| **Tr*[4]** | 268 | 1987 | 368 | 1.4 | 117.2 | 6.0 | 87.1 |
| **Tr*[6]** | 310 | 1989 | 443 | 1.8 | 429.9 | 7.9 | 45.8 |
| **Tr*[8]** | 347 | 1957 | 495 | 2.2 | 487.1 | 4.3 | 59.4 |
| **Tr*[10]** | 376 | 1894 | 508 | 2.3 | 751.8 | 7.7 | 54.3 |
| **Glu[4]** | 242 | 1968 | 347 | n.d. | n.d. | n.d. | n.d. |
| **Glu[6]** | 292 | 1985 | 430 | 8.8 | 307.3 | n.d. | n.d. |
| **Glu[8]** | 334 | 1907 | 487 | 4.4 | 391.8 | n.d. | n.d. |
| **Glu[10]** | 367 | 1822 | 523 | 3.2 | 469.6 | n.d. | n.d. |
| **Glu[12]** | 393 | 1803 | 564 | 2.7 | 1106.4 | n.d. | n.d. |

Data determined for **Tr*[n]** with $n = 4$–10 (in hexane) and **Glu[n]** with $n = 4$–12 (in DCM), as well as the time constants $\tau_1$ ($S_n \rightarrow S_1$), $\tau_2$ ($S_1 \rightarrow T_1$), and $\tau_3$ ($T_1 \rightarrow S_0$) obtained by global analysis of the transient absorption data using a convolution of the instrument response function and three exponential functions (*n.d.* not determined). Source data are provided in the Source Data file.

quantum yields on the order of only $\Phi_F = 10^{-4}$ and negligible differences of the fluorescence quantum yields with increasing conjugation length (Supplementary Table 2). Because of the low quantum yields, multiple attempts to determine the fluorescence lifetime by means of either single photon counting or up-conversion were unsuccessful. The low quantum yields are not surprising given that fluorescence does not occur from the lowest electronic state, and rapid deactivation pathways such as internal conversion to the non-emissive $S_1$ state as well as intersystem crossing presumably contribute to a lowering of the $S_n \rightarrow S_0$ fluorescence quantum yields (vide infra).

**Time-resolved spectroscopy.** Time-resolved absorption spectroscopy with **Glu[n]** ($n = 6, 8, 10, 12$; in DCM) and **Tr*[n]** ($n = 4, 6, 8, 10$; in hexane) at excitation wavelengths of 390 and 258 nm, respectively, show very similar spectroscopic features (Fig. 3, Supplementary Fig. 4–11) and dynamics (Table 1) for all molecules. The excitation wavelengths have been chosen to pump into either $\lambda_{main}$ or $\lambda_{weak}$, and Supplementary Fig. 12 illustrates the lack of any meaningful dependence of the observed species and lifetimes on the excitation wavelength. For **Glu[12]** as a representative example, a broad photoinduced absorption (PIA) centered at 586 nm appears promptly after excitation (Fig. 3a). Within the first few picoseconds, this broad PIA narrows substantially, and a vibrational progression appears. Simultaneously, the center of the PIA undergoes a blue shift to 577 nm. Both effects can be related to the relaxation to $S_1$, consistent with the discussion of the steady-state UV-Vis and fluorescence spectra. On a longer timescale, we observe the rise of a second PIA at 464 nm, assigned to triplet-triplet absorptions $T_1 \rightarrow T_n$ upon intersystem crossing.

The underlying dynamics are well reproduced by means of deconvoluting the instrument response function and three exponential functions. We interpret the three time constants to describe the ultrafast transformation of $S_n/S_{2/3}$ into $S_1$, the intersystem crossing of the latter to yield $T_1$, and the subsequent decay to the ground state $S_0$. The assignment of the intermediate, evolution-associated spectroscopic features to the $S_1$ state are corroborated by the fact that the population of the state with time constants between 1 and 10 ps is, by far, too slow for a direct vertical excitation, which typically occurs within the instrument response function of the laser system (<1 ps). Moreover, previous studies of related systems by means of time-resolved infrared spectroscopy excluded processes such as intramolecular/vibrational relaxation and/or solvation processes[35].

Comparing the transient absorption spectra of **Tr*[10]** and **Glu[10]** indicates that the end groups have no major impact on these photo-initiated processes (Fig. 3b, c). In both series of oligoynes, the deactivation of $S_n$ as well as $S_{2/3}$ and simultaneous population of $S_1$ is followed by intersystem crossing to $T_1$, which slows with increasing length of the oligoyne segment (and is slightly faster for **Glu[n]** compared with **Tr*[n]**, possibly because of the higher excitation energy used in the latter series). The rapid transformation of the excited state ($S_n$) into a non-emissive, lower energy state ($S_1$) is likely to be nearly quantitative, a fact that is derived from the very low quantum yields of the competing fluorescence pathway (vide supra).

For **Glu[8]** and **Glu[6]** an additional broad peak is discernable as part of the $S_1 \rightarrow S_n$ transition that is subject to a slight blue-shift during intersystem crossing (Supplementary Figs. 10–11). Fazzi et al.[34] have reported similar observations in their analysis of α,ω-dinaphthyloligoynes and attributed these peaks to the superposition of several transitions from the first singlet excited state. The observation of these $S_1 \rightarrow S_n$ transitions corroborates that the excitation of the shorter oligoynes at a lower energy than their most intense transition $\lambda_{main}$ furnishes exactly the same excited state behavior as for the longer derivatives that are excited within $\lambda_{main}$. This, in turn, lends further support to our hypothesis that the main optical transition $\lambda_{main}$ is not related the $S_0 \rightarrow S_1$ transition. Instead, a higher, short-lived $S_n$ excited state is populated by the pump pulse, which is rapidly deactivated to the $S_1$ state, which is then probed by transient absorption spectroscopy.

The same final, evolution-associated spectroscopic features assigned to the $T_1 \rightarrow T_n$ absorptions in the femtosecond transient absorption time window are also discernable by nanosecond spectroscopy, which we have performed in the absence and presence of oxygen (Supplementary Fig. 13–16). As a representative example, the differential absorption spectrum of **Tr*[10]** shows maxima at 440, 500, 560, and 600 nm, which have lifetimes of 7.7 µs in argon and 54 ns in oxygen-saturated solutions (Table 1). This difference in lifetimes indeed confirms that these features are related to a triplet excited state for which deactivation occurs via triplet-triplet energy transfer to molecular oxygen. The formation of singlet oxygen was independently confirmed in emission experiments, which show the characteristic signal at 1275 nm (Supplementary Fig. 17).

Both steady-state and time-resolved absorption spectroscopy hence provide unambiguous experimental evidence that the observed weak intensity absorptions at higher wavelengths, $\lambda_{weak}$, are intrinsic photophysical features of oligoynes.

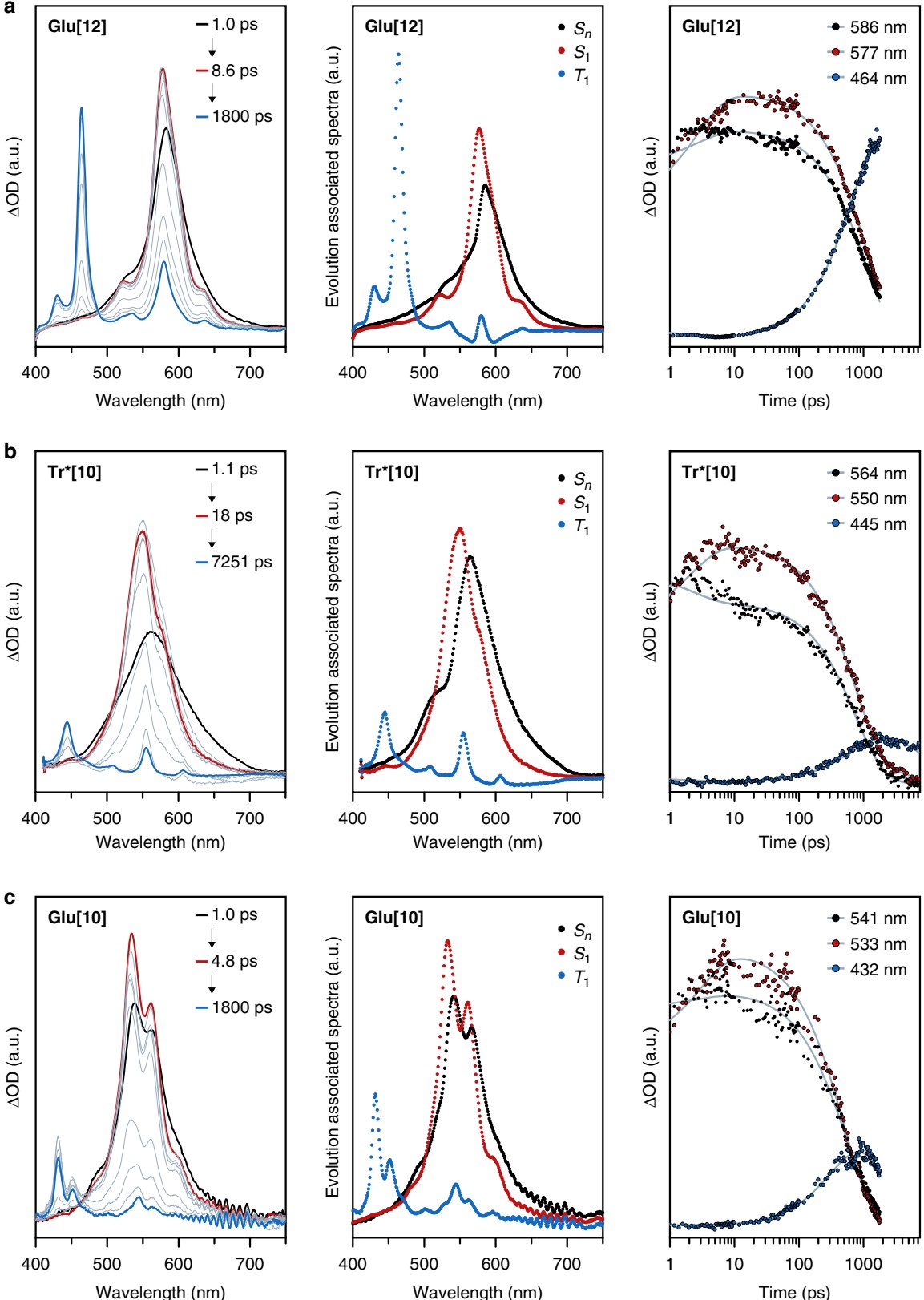

**Fig. 3 Transient absorption spectroscopy and analysis.** Selected transient absorption spectra (first column), evolution-associated spectra obtained from global analysis (second column), and characteristic dynamics with a fit to the corresponding data (gray line, third column) of **a Glu[12]** (DCM), **b Tr*[10]** (hexane), and **c Glu[10]** (DCM). Source data are provided as a Source Data file.

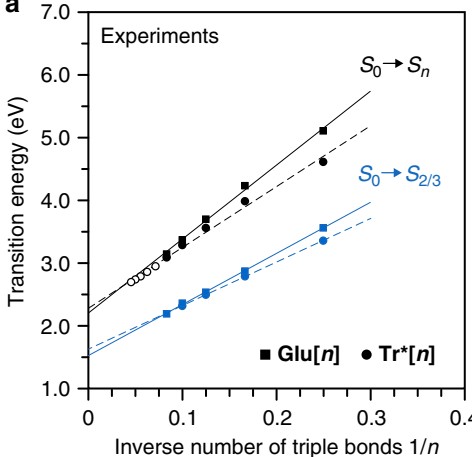

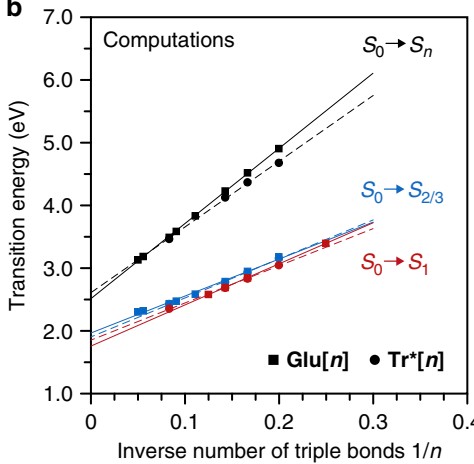

**Fig. 4 Extrapolation of experimental and computational transition energies. a** Plot of the energies of the experimentally determined $S_0 \rightarrow S_n$ (main) transitions and the $S_0 \rightarrow S_{2/3}$ (weakly allowed) transitions against the inverse number of carbon–carbon triple bonds $1/n$ for **Glu[n]** in DCM and **Tr*[n]** in hexane (open symbols: values for **Tr*[n]** | from the literature)[17]. **b** Plot of the energies of the computationally determined $S_0 \rightarrow S_n$ (main), $S_0 \rightarrow S_{2/3}$ (weakly allowed), as well as $S_0 \rightarrow S_1$ (forbidden) transitions against the inverse number of carbon–carbon triple bonds $1/n$ for **Glu[n]** in DCM and **Tr*[n]** in hexane. The saturation values have been determined by linear extrapolation to the **Glu[n]** data (solid lines) and the **Tr*[n]** data (dashed lines). Source data are provided in the Source Data file.

**Computational analysis of the optical transitions.** The tentative assignment of the main ($\lambda_{main}$) and weakly allowed ($\lambda_{weak}$) optical absorptions of the two series of oligoynes **Glu[n]** and **Tr*[n]** to $S_0 \rightarrow S_n$ and $S_0 \rightarrow S_{2/3}$ transitions, respectively, is corroborated by plotting the experimentally and computationally determined energies against the inverse number of carbon–carbon triple bonds $n$ (Fig. 4). Both series each converge to virtually identical saturation values. The linear extrapolation of the $S_0 \rightarrow S_n$ transition energies ($\lambda_{main}$) for $1/n \rightarrow 0$ furnishes saturation values for the experimental optical transitions of the **Glu[n]** and **Tr*[n]** series of $E_{opt} = 2.21$ eV ($\lambda_{opt} = 561$ nm) and 2.28 eV (543 nm), respectively. These values compare very well to previously determined values of, e.g., $E_{opt} = 2.18$–2.19 eV ($\lambda_{opt} = 565$–570 nm) for $^iPr_3Si$-terminated oligoynes[15], oligoynes with third-generation Frechet-type dendrons as end groups[16], or oligoynes endcapped with rhenium complexes[18]. An extrapolation of the significantly red-shifted $S_0 \rightarrow S_{2/3}$ transition energies ($\lambda_{weak}$) for $1/n \rightarrow 0$,

however, results in saturation values of $E_{opt} = 1.53$ eV ($\lambda_{opt} = 810$ nm) for the **Glu[n]** series and $E_{opt} = 1.64$ eV ($\lambda_{opt} = 756$ nm) for the **Tr*[n]** series, which is ~0.6–0.7 eV lower than the $E_{opt}$ value suggest by analysis of $\lambda_{main}$ across the entire series of **Tr*[n]** ($n = 2$–22)[17].

In good agreement with the experimental results, saturation values for the **Glu[n]** and **Tr*[n]** series of 2.59 eV (478 nm) and 2.50 eV (496 nm), respectively, are obtained from the computations of the $S_0 \rightarrow S_n$ transitions ($\lambda_{main}$), as well as 1.89 eV (656 nm) and 1.96 eV (634 nm) for the and $S_0 \rightarrow S_{2/3}$ transitions ($\lambda_{weak}$), respectively. All trends and differences are hence fully reproduced, although the energies are slightly overestimated. Although the employed CAM-B3LYP functional provides good estimates for the excited state of $\pi$-conjugated chains[52], it is known to have a tendency of over-localizing the electron density distribution[52,53] and, in turn, overestimating transition energies, especially when the effects of vibronic coupling are neglected[54]. This is also the reason why the computations do not fully reproduce the minor differences between the **Glu[n]** and **Tr*[n]** series and the deviations from the linear extrapolation for short oligomers that can be attributed to hyperconjugation to the end groups in the **Tr*[n]** series (Supplementary Figs. 18 and 19)[49].

The good agreement between experimental and computational results for both series and both the $S_0 \rightarrow S_n$ (main) transitions as well as the $S_0 \rightarrow S_{2/3}$ is relevant because it gives confidence in the computational determination of the $S_0 \rightarrow S_1$ transitions that cannot be probed experimentally (Fig. 4b). Notably, the observed deviations are smaller for the $S_0 \rightarrow S_{2/3}$ (weakly allowed) transitions because the quasi-forbidden state is significantly more compact than the bright state and does not benefit as much from a delocalization by hyperconjugation toward the end groups in the **Tr*[n]** series.

Using the values of $\lambda_{weak}$ therefore suggests that the optical gap of oligoynes of finite length is significantly smaller than previously reported, converging to $E_{opt} = 1.53$ eV for the **Glu[n]** series and $E_{opt} = 1.64$ eV for the **Tr*[n]** series. As the $S_0 \rightarrow S_{2/3}$ transitions should become symmetry-forbidden for a sufficiently large number of triple bonds in $(C \equiv C)_n$, it may be debatable whether this estimate also holds true for carbyne. In practice, however, even minor deviations from an idealized linear geometry will render $S_0 \rightarrow S_{2/3}$ transitions partially allowed. The photophysical properties of carbyne, and its photochemistry, are therefore better described by the optical gap estimated based on extrapolations of the data presented herein using $\lambda_{weak}$, i.e., $E_{opt} = 1.5$–1.6 eV, about 0.6–0.7 eV smaller than previously inferred.

Moreover, the presented results provide an estimate for the fundamental gap of carbyne ($E_g$), which should be below the $S_0 \rightarrow S_{2/3}$ but above the $S_0 \rightarrow S_1$ transition energy (owing to the exciton binding energy). We conclude that its upper boundary should hence be $E_g = 1.6$ eV, as well. A lower boundary can be estimated from the extrapolation of the computational determination of the $S_0 \rightarrow S_1$ transition energies for $1/n \rightarrow 0$, which furnishes saturation values of 1.75 and 1.84 eV for the **Glu[n]** and **Tr*[n]** series, that is, ~0.2 eV below the computationally determined saturation values for the $S_0 \rightarrow S_{2/3}$ transition of 1.9–2.0 eV (at the given DFT level). It should be noted that the more accurate prediction method for saturation, reported by Meier et al.[55], has also been attempted but is unreliable owing to the limited number of experimental data points. Given the systematic and explicable overestimation of all transition energies of ~0.3 eV across both $S_0 \rightarrow S_{2/3}$ and $S_0 \rightarrow S_n$ transitions and both series of molecules in the computations as compared to the respective experimental values, the lower boundary of the fundamental gap should hence be estimated to be on the order of $E_g = 1.4$–1.5 eV.

A systematic investigation of the ground-state and excited-state properties of two series of oligoynes has allowed us to determine the intrinsic photophysical properties of oligoynes, which are of particular interest as analogs for the elusive carbon allotrope carbyne. Our findings prove that the series of "weak absorptions", $\lambda_{weak}$, that one can observe at significantly lower transition energies than the main optical absorptions, $\lambda_{main}$, are intrinsic photophysical features of oligoynes. Combining the experimental and computational results enabled us to assign $\lambda_{weak}$ to the weakly allowed $S_0 \rightarrow S_{2/3}$ transition to the degenerate state $E$. The main optical absorptions hence correspond to a higher-energy $S_0 \rightarrow S_n$ transition, whereas the lowest $S_0 \rightarrow S_1$ transition remains dipole-forbidden. Moreover, our results require that both the optical gap ($E_{opt}$) and the fundamental gap ($E_g$) of oligoynes and carbyne are reassessed. We find that the optical gap of oligoynes converges to $E_{opt} = 1.5$–$1.6$ eV. These values are $0.6$–$0.7$ eV smaller than previous estimates based on $\lambda_{main}$. Furthermore, the combined experimental and computational results provide an estimate of $1.4$–$1.6$ eV for the fundamental gap ($E_g$) of a hypothetical $sp$-hybridized carbon allotrope carbyne. This value compares to ca. $5.5$ eV in diamond ($sp^3$)[56], $8.8$ eV for polyethylene ($CH_2$–$CH_2$)$_n$ ($sp^3$)[57], and $1.8$ eV in polyacetylene ($CH$=$CH$)$_n$ ($sp^2$)[44]. The results of the present study provide more accurate and experimentally verified predictions for the photophysical properties of molecules with extended $\pi$-conjugated systems based on $sp$-hybridized carbons that will help guide the preparation of carbon-rich materials with tailored properties in the future.

## Methods

**Spectroscopy**. All used solvents were purchased from commercial suppliers and used without further purification. The synthesis of the two series of oligoynes **Tr**\*[**n**] and **Glu**[**n**] with $n = 2, 6$ has been previously reported[17,22], and the compounds **Glu**[**n**] with $n = 4, 8, 10, 12$ were prepared following these procedures (see Supplementary Method 2 for details). Steady-state UV-Vis absorption spectra were acquired at room temperature (rt) in solution using a Perkin Elmer Lambda 2 spectrometer (**Tr**\*[**n**] series) and a JASCO V670 spectrometer (**Glu**[**n**] series). Steady-state fluorescence spectra were measured at rt in solution with a Horiba FluoroMax3 spectrometer in the visible detection range and a Horiba FluoroLog3 spectrometer in the near-infrared detection range. Femtosecond transient absorption experiments (rt/argon-saturated solution) were carried out with amplified Ti:Sapphire femtosecond laser systems from Clark MXR (**Tr**\*[**n**] series: CPA-2101, fundamental 775 nm, 1050 Hz; **Glu**[**n**] series: CPA-2001, fundamental 780 nm, 1000 Hz) using transient absorption pump/probe detection systems. The 258 nm excitation wavelength was generated via third harmonic generation of the 775 nm CPA-2101 output. The excitation pulses at 390 nm were generated via frequency doubling (second harmonic generation) of the 780 nm fundamental of the CPA-2001. Nanosecond transient absorption experiments were carried out with a non-commercial setup built at Friedrich-Alexander University Erlangen-Nürnberg. The samples were measured in argon-saturated solution at rt to prevent degradation and quenching by oxygen unless stated otherwise. As white light source, a pulsed 450 W XBO lamp (Osram) was used. The 266 nm excitation was generated via fourth harmonic generation of the 1064 nm laser output of a Nd: YAG laser system (Brilliand-Quantel, 5 ns pulse width). The transient absorption data were fitted by means of global as well as multiwavelengths analyses. The routines for these analyses are described in detail in Supplementary Method 1.

**Computations**. Computations using a range-separated hybrid functional (e.g., CAM-B3LYP) generally provide a more reliable description of the spectroscopic properties of oligoynes (e.g., energies and hyperpolarizability) than standard semi-local or global hybrid approximations with increasing chain length[58]. All the oligoyne geometries were optimized at the CAM-B3LYP/6-31 G(d) level[59–61] in Gaussian09[62]. The first 40 singlet excitations were computed using linear-response time-dependent functional theory at the TD-CAM-B3LYP/6-31 + G(d) level. For both the geometry optimization and vertical excitations, solvation was accounted for by using the PCM polarizable continuum model for hexane ($\varepsilon = 1.9$) or dichloromethane ($\varepsilon = 8.9$) as implemented in Gaussian09[63–66]. Following the definition proposed by Gieseking et al.[67], bond length alternations were computed as the difference between the mean lengths of the carbon–carbon single-like bonds and carbon–carbon triple-like bonds. We here consider the $D_{\infty h}$ parent compound **H**[**n**] ($n = 4, 6, 8, 10, 12$) as well as the two endcapped oligoyne series, i.e., the $C_2$-symmetric **Glu**[**n**] ($n = 6, 8, 10, 12$) and the $D_3$-symmetric **Tr**\*[**n**] ($n = 4, 6, 8, 10$). To facilitate the computations, the terminal groups were truncated by replacing the *tert*-butyl moieties of **Tr**\*[**n**] and the OAc substituents of **Glu**[**n**] with protons.

## Data availability

The data that support the findings of this study and the source data underlying Figs. 2–4, Table 1, Supplementary Figs. 1–17, and Supplementary Tables 1–2 are provided as a Source Data file. The data are also available from the corresponding authors upon reasonable request.

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

## Acknowledgements

This work was supported by the Deutsche Forschungsgemeinschaft (DFG) as part of SFB 953 "Synthetic Carbon Allotropes" and the Natural Sciences and Engineering Research Council of Canada (NSERC). The Graduate School Molecular Science (GSMS) of the FAU Erlangen-Nürnberg is gratefully acknowledged for its generous support. J.C.B. thanks Professor Jacques-Edouard Moser for access to the femtosecond pump-probe setup. C.C. and L.V. thank the Swiss National Science Foundation for funding (SNF Grant 156001) and C.C. thanks Dr. Stephan Steinmann for insightful discussion. H.F. and S.S. acknowledge funding from the European Research Council (ERC Grant 239831).

## Author contributions

S.S., E.C., and E.J. synthesized oligoynes and performed steady-state absorption spectroscopy under the supervision of R.T. and H.F. J.Z. and J.C.B. performed steady-state absorption, fluorescence, as well as time-resolved spectroscopy under the supervision of D.G. and H.F. L.V. and E.B. furnished the computational analyses under the supervision of C.C. J.Z., S.S., and J.C.B. wrote the manuscript under the supervision of D.G., C.C., R.T., and H.F.

## Competing interests

The authors declare no competing interests.
