## [Peer Review File · Nature Communications]

Reviewers' comments:

Reviewer #1 (Remarks to the Author):

In this work (NCMMS-20-03672-T), the authors investigated the optical transitions of a series of oligynes by systematic analyses of photophysical properties in combination with computational approach. The results, to my viewpoint, are convincing, which may gain credit in fundamental due to its correction on the previous erroneous assignment of oligynes and carbynes. However, I cannot but have certain dilemma on the importance regarding its broader interest to the scientific community. Despite the comprehensive study in approving the new assignment of the optical gap, throughout the text I do not see relevant statements to elaborate if these findings could constitute a significant advance in future progress either in fundamental or applications. From this viewpoint, unless this finding has great potential extension in fundamental or applications, otherwise, I lean my decision toward its suitability for publication in a specialized physical or physical chemistry oriented journals. Other comments are listed below.

1. The authors successfully measured the fluorescence spectra of Tr*[n] series and attributed the fluorescence to be $S_n \rightarrow S_0$ transition. However, the corresponding time constants and assignments in TA spectra are not taken into account in the manuscript. This makes the assignment of transitions deduced from TA spectra unconvincing. Authors should address this part.

2. What is the T1 to S0 decay time constant in the case of Glu[n] series? The rise of triplet-triplet absorptions has been assigned in the text, there is no reason that authors are not able to resolve the population decay rate of T1.

Minor point:

1. "n.d." wasn't defined in either Table 1 or text.

Reviewer #2 (Remarks to the Author):

The authors reassess the electronic structure and optical absorption energies of oligynes, based on optical spectroscopy assisted by calculations. Their main finding, which is quite convincing in light of the evidence given, is that both optical and fundamental gaps are substantially lower than previously believed. This revision of a long-held belief is already reason enough to warrant publication in Nature Communications. I am therefore pleased to recommend this article for publication.

I would suggest that the authors consider the following minor points of revision:

1. The caption of Figure 1 refers to panels (a) through (e) but the figure itself has only (a) to (c). It also refers to the chemical structures last when they are first in the figure. Please correct.
2. While Bredas has indeed emphasized the distinction between the fundamental and the optical gap in the context of organic electronics, it has long been known in the solid state physics literature, e.g. in the Reviews of Modern Physics articles by Onida et al. [RMP 74, 601 (2002)] or Kummel et al. [RMP 80, 3 (2008)].
3. Unless I missed it, the text doesn't refer anywhere to the experimental and computational details given in the Supplementary Information. This is unfortunate, as from time to time these details are mentioned in the main text where relevant. At the very least reference to the Supplementary Information should be added before any mention of results. I would also encourage the authors to seriously consider a very short section in the main text that describes the essential details, with the rest indeed given in the Supplementary Information.

Reviewer #3 (Remarks to the Author):

In the manuscript "Optical gap and fundamental gap of oligoynes and carbyne" the authors examine two sets of oligoynes in order to find the HOMO-LUMO energy difference for polyynes. I agree with the authors that finding the optoelectronic properties of carbon allotropes is of fundamental importance, but I very much dislike the authors' mixing of solid state physics (band gap) and molecular spectroscopy (states) of nomenclature, which makes the manuscript very hard to read. Below are detailed remarks.

1. page 2: "...weak bands are observed at higher wavelengths than the λ_{main} bands in solution measurements..." It is hard to know when the discussion is on molecules in solution or solid state. Please clearly specify throughout the manuscript.

2. Page 2: References 29-31 use H capped oligoynes. Please describe the work and conclusions from this earlier work more thorough. This because the molecules used are very similar to your own, and a differentiation between earlier work and this work needs to be stated in order to assess novelty.

3. Page 2: The two oligoynes series are claimed to have non-conjugated end-groups. This is true (formally), but the photophysics differs considerable (the absorption maximum changes 10-30 nm)! Furthermore, the electron density difference between the ground and excited state (figure s16) differs considerable between the two series. The electron density spreads out on the endcapping groups for the Tr series but not for the glu series. An analysis regarding these things is in order. I am aware that the authors say that the phenomena is "likely be attributed to hyperconjugation" on page 7. But this is a statement without any experimental or theoretical support.

4. Figure 1. The caption refers to "d" and "e", but the figure itself is only a-c. Also, in the text a more thorough discussion that relates the absorption of the weak/disallowed low energy transitions to point group reasoning would be interesting.

5. Figure 4: "...distinguish between the "optical gap", E_{opt} and the "fundamental gap", E_g (Figure 1b,c)". Figure 1 shows an energy diagram having the state notation. Furthermore, in the next paragraph (which I really like) the notation goes back to using states. At minimum, a through explanation in the text what is meant with each term is in order. Another issue I have with the phrase, fundamental gap, is where triplet states fits in. The conclusion of the manuscript is that the fundamental energy gap is about 1.6 eV, but the energy of the triplet state is less than 1 eV (based on phosphorescence!

6. Figure 2c. Why is not the fluorescence scanned to higher energies for TR6 and TR8 (as it is done with Tr10)? I think information is missing because of this and cause the information on the Stokes shift in table S2 to be erroneous. Furthermore, please state excitation wavelength and detection wavelength in the caption.

7. Page 7: "reflecting the larger degree of conjugation in longer oligoynes", can the authors explain why. The IR signal can go to lower energies either because an increase in effective mass of the vibrating unit, or a decrease in binding energy. As I see it, if something is reflecting a higher degree of conjugation, then two different vibrations (the carbon-carbon single bond and the triple bond) would gradually merge to one single vibration.

8. Page 8: “from the longest wavelength absorption and shortest wavelength fluorescence...” The authors show that the forbidden $S_0 \rightarrow S_2-3$ state do absorb light, therefore is this statement erroneous.

9. Page 9: Why was the two series excited at different wavelength in the fs TA experiments? Was there a rational for the excitation wavelength? Furthermore, can the forbidden $S_0 \rightarrow S_1$ be excited directly? A lack of the initial decay time when analysing a decay after direct excitation would significantly strengthen the mechanism suggested in the TA measurements.

Submission of Revised Manuscript NCOMMS-20-03672-T – Response to Referee Comments

Response to Reviewer 1:

Reviewer comment:

In this work (NCOMMS-20-03672-T), the authors investigated the optical transitions of a series of oligoynes by systematic analyses of photophysical properties in combination with computational approach. The results, to my viewpoint, are convincing, which may gain credit in fundamental due to its correction on the previous erroneous assignment of oligoynes and carbyne.

Response to Reviewer comment:

We would like to thank the reviewer for the concise summary of our work and his/her very positive assessment of the original manuscript. We sincerely hope that the answers provided below as well as the revised manuscript meet the remaining concerns.

Reviewer comments:

However, I cannot but have certain dilemma on the importance regarding its broader interest to the scientific community. Despite the comprehensive study in approving the new assignment of the optical gap, throughout the text I do not see relevant statements to elaborate if these findings could constitute a significant advance in future progress either in fundamental or applications. From this viewpoint, unless this finding has great potential extension in fundamental or applications, otherwise, I lean my decision toward its suitability for publication in a specialized physical or physical chemistry oriented journals.

Response to Reviewer comment:

The synthesis of new types of carbon-rich molecules and materials with extended π -conjugated systems is a topic of continued, fundamental interest, and their characterization often relies heavily on spectroscopic methods. We hope that the results presented in this manuscript provide a fundamental understanding of the correlation between the photophysical characteristics and the properties of such molecules. In the case of oligoynes, both theory and selected experimental examples of studies have appeared and offered an interpretation of specific examples. The narrow scope of earlier studies has, however, failed to offer a broad explanation to guide the interpretation of the absorption spectra of oligoynes (including, for example, the correct identification of the HOMO-LUMO energy gap). The preparation of new carbon-rich structures with tailored properties and eventual applications depends on accurate, general predictions which is what we provide in this manuscript. We believe that we have outlined this to the best of our ability in the introduction.

Reviewer comment:

The authors successfully measured the fluorescence spectra of $\text{Tr}^*[n]$ series and attributed the fluorescence to be $S_n \rightarrow S_0$ transitions. However, the corresponding time constants and assignments in TA spectra are not taken into account in the manuscript. This makes the assignment of transitions deduced from TA spectra unconvincing. Authors should address this part.

Response to Reviewer comment:

We thank the Reviewer for bringing this aspect to our attention. As a matter of fact, we attribute the fluorescence to the $S_n \rightarrow S_0$ transitions as they mirror the $S_0 \rightarrow S_n$ absorptions. We wish to

point out that the fluorescence lifetime determination by means of single photon counting was unsuccessful, despite numerous attempts. Considering, however, fluorescence quantum yields (Φ_F) on the order of 10^{-4} , internal conversion to afford the non-emissive S_1 , which proceeds within several picoseconds, outcompetes the fluorescent deactivation by far. In light of the aforementioned, the time constants relating to fluorescence play no major role within the timeframe of the transient absorption spectra and analyses thereof. We have edited the corresponding section in the revised version of the manuscript to further clarify these points (pages 9 and 12).

Reviewer comment:

What is the T_1 to S_0 decay time constant in the case of **Glu[n]** series? The rise of triplet-triplet absorptions has been assigned in the text, there is no reason that authors are not able to resolve the population decay rate of T_1

Response to Reviewer comments:

We agree with the Reviewer that it is possible to resolve the T_1 decay time constants in the case of the **Glu[n]** series, as the triplet-triplet absorptions are discernable in the corresponding fs-resolved transient absorption spectra. We wish to emphasize, however, that we describe in the *Supplementary Information* (3. General Procedures and Methods), that the transient absorption data were collected independently in two different labs for the **Tr*[n]** and the **Glu[n]** series, respectively. For measurements regarding the **Tr*[n]** series, a non-commercial nanosecond transient absorption setup, built at the Friedrich-Alexander-University, was used. For the **Glu[n]** series such a set-up was, however, unavailable. By virtue of the spectral similarities between the **Tr*[n]** and the **Glu[n]** series, we are confident that establishing the T_1 decay time constants for the **Glu[n]** series is unlikely to provide new mechanistic insights.

Reviewer comment:

Minor point: "n.d." wasn't defined in either **Table 1** or text.

Response to Reviewer comment:

We thank the reviewer for pointing this out and we have added the definition for "n.d." as "not determined" to the header of **Table 1** in the revised version of the manuscript.

Response to Reviewer 2:

Reviewer comment:

The authors reassess the electronic structure and optical absorption energies of oligoynes, based on optical spectroscopy assisted by calculations. Their main finding, which is quite convincing in light of the evidence given, is that both optical and fundamental gaps are substantially lower than previously believed. This revision of a long-held belief is already reason enough to warrant publication in Nature Communications. I am therefore pleased to recommend this article for publication.

Response to Reviewer comment:

We are very grateful for the reviewer's positive evaluation of our original manuscript and for acknowledging the significance of our findings. We would like to thank him/her for the helpful suggestions. We sincerely hope that our point-by-point response as well as the revised manuscript will resolve any loose ends.

Reviewer comment:

The caption of **Figure 1** refers to panels (a) through (e) but the figure itself has only (a) to (c). It also refers to the chemical structures last when they are first in the figure. Please correct.

Response to Reviewer comment:

We thank the reviewer for pointing us to this mistake. We have corrected the Figure Caption of **Figure 1** accordingly.

Reviewer comment:

While Bredas has indeed emphasized the distinction between the fundamental and the optical gap in the context of organic electronics, it has long been known in the solid state physics literature, e.g. in the Reviews of Modern Physics articles by Onida et al. [RMP 74, 601 (2002)] or Kummel et al. [RMP 80, 3 (2008)].

Response to Reviewer comment:

We would like to thank the reviewer for pointing us to these relevant additional references, which we have added as references 46 and 47 to the revised version of the manuscript.

Reviewer comment:

Unless I missed it, the text doesn't refer anywhere to the experimental and computational details given in the *Supplementary Information*. This is unfortunate, as from time to time these details are mentioned in the main text where relevant. At the very least reference to the *Supplementary Information* should be added before any mention of results. I would also encourage the authors to seriously consider a very short section in the main text that describes the essential details, with the rest indeed given in the *Supplementary Information*.

Response to Reviewer comment:

We thank the reviewer for raising this point. We have added a "Methods" section to the revised version of the manuscript, which includes a description of the important experimental procedures and a description of the computation details. Moreover, we have added some cross-references to the *Supplementary Information* and the Methods section to the revised version of the manuscript.

Response to Reviewer 3:

Reviewer comment:

In the manuscript "Optical gap and fundamental gap of oligoynes and carbyne" the authors examine two sets of oligoynes in order to find the HOMO-LUMO energy difference for polyynes. I agree with the authors that finding the optoelectronic properties of carbon allotropes is of fundamental importance. I very much dislike the authors' mixing of solid-state physics (band gap) and molecular spectroscopy (states) of nomenclature, which makes the manuscript very hard to read. Below are detailed remarks.

Response to Reviewer comment:

We thank the reviewer for her/his concise summary of our work and for referring to its fundamental importance. We acknowledge his/her criticism concerning our use of a "mixed" terminology. The notion of a "band gap" is frequently used in the area of conjugated organic molecules and polymers, where experimental data for molecular building blocks are often directly correlated with the underlying physical processes. In sound agreement with the outstanding article by Bredas (ref. 45; *Mater. Horiz.* **2013**, *1*, 17–19), in which important distinctions are made and related discussions in the physics literature are highlighted (new refs. 46 and 47; *Rev. Mod. Phys.* **2002**, *74*, 601–659; *Rev. Mod. Phys.* **2008**, *80*, 3–60), we wish to focus on this particular aspect in the context of oligoynes and their experimental investigations. We hope that our revisions and our point-by-point responses address all remaining concerns of the reviewer.

Reviewer comment:

Page 2: "...weak bands are observed at higher wavelengths than the λ_{main} bands in solution measurements..." It is hard to know when the discussion is on molecules in solution or solid state. Please clearly specify throughout the manuscript.

Response to Reviewer comment:

In the present study, *all* spectroscopy experiments were carried out in solution. In the revised version of the manuscript we have underlined this aspect on multiple occasions. Moreover, we have added a Methods section to the revised version of the manuscript that further describes the detailed experimental approach.

Reviewer comment:

References 29-31 use H capped oligoynes. Please describe the work and conclusions from this earlier work more thorough. This because the molecules used are very similar to your own, and a differentiation between earlier work and this work needs to be stated in order to assess novelty.

Response to Reviewer comment:

The parent series, **H[n]**, is unique since there is no influence of the end groups on the electronic/physical properties regardless of length. Moreover, the instability of derivatives longer than ca. **H[3]** (see for example Armitage *et al.* *J. Chem. Soc.* **1952**, 2010–2014), significantly limits the relevance of their comparison to larger series of "end-capped" oligoynes. Consequently, we now present a systematic experimental study of "end-capped" oligoynes that allows for an extrapolation of the observed transition towards carbyne. We have added the following statement to the manuscript to highlight the fundamental, albeit privileged characteristics of the parent **H[n]** series (page 2): "Characteristics of the **H[n]** series, while foundational to the present study, are unique due to the lack of terminal substituents which results in $D_{\infty h}$ symmetry for all members of the series irrespective of length."

Reviewer comment:

The two oligoyne series are claimed to have non-conjugated end-groups. This is true (formally), but the photophysics differs considerable (the absorption maximum changes 10-30 nm)! Furthermore, the electron density difference between the ground and excited state (**Figure S16**) differs considerable between the two series. The electron density spreads out on the endcapping groups for the **Tr[n]** series but not for the **Glu[n]** series. An analysis regarding these things is in order. I am aware that the authors say that the phenomena is “likely be attributed to hyperconjugation” on page 7. But this is a statement without any experimental or theoretical support

Response to Reviewer comment:

The influence of non-conjugated end groups (*i.e.*, those composed of sp^3 -hybridized atoms) is well documented in the literature and the effects of the end groups diminishes rapidly as a function of length (new reference 49 added to the revised manuscript; Agarwal, N. R. *et al. J. Raman Spectrosc.* **44**, 1398–1410 (2013)). Indeed, for compounds that have the length of a decayne (*i.e.*, **Tr[10]** vs **Glu[10]**) the difference is only ca. 0.08 eV (3.30 vs 3.38 eV). Moreover, the assignment to hyperconjugation is supported by the computational analysis that was carried out for the two series, which shows orbital overlap between the π system and the end group in particular for the **Tr[n]** derivatives (see Supplementary Figures S18 and S19). The revised version of the manuscript has been carefully edited to point to these aspects in support of hyperconjugation.

Reviewer comment:

The caption refers to “d” and “e”, but the figure itself is only a-c. Also, in the text a more thorough discussion that relates the absorption of the weak/disallowed low energy transitions to point group reasoning would be interesting.

Response to Reviewer comment:

We would like to thank the reviewer for pointing us to the mistake in the caption, which we have corrected in the revised manuscript. A full discussion of using point group reasoning would likely be interesting to some readers, but beyond the grasp of the majority. Since an analogous discussion exists in the literature (ref. 31, Wakabayashi *et al. J. Phys.: Conf. Ser.* **428**, 012004 (2013)). We have added a statement to the revised version of the manuscript to direct interested readers to the appropriate work (see page 5): “(see Wakabayashi *et al.* for a detailed analysis of the relevant point groups)”.

Reviewer comment:

Figure 4: “...distinguish between the “optical gap”, E_{opt} and the “fundamental gap”, E_g (**Figure 1b,c**)”. **Figure 1** shows an energy diagram having the state notation. Furthermore, in the next paragraph (which I really like) the notation goes back to using states. At minimum, a thorough explanation in the text what is meant with each term is in order. Another issue I have with the phrase, fundamental gap, is where triplet states fits in. The conclusion of the manuscript is that the fundamental energy gap is about 1.6 eV, but the energy of the triplet state is less than 1 eV (based on phosphorescence)!

Response to Reviewer comment:

We would like to point out that the use of the state terminology is restricted to the optical transitions. This aspect has been clarified in the text (page 4). Moreover, we respectfully disagree with the conclusion of Reviewer 3 regarding the energy of the triplet excited state. The emission spectrum depicted in **Figure S15** is that of singlet oxygen generated by **Tr*[10]** measured in oxygen saturated hexane. In these experiments, **Tr*[10]** acts as photosensitizer,

which uses its energy after light absorption at 315 nm to convert oxygen in its non-emissive triplet ground state to its emissive, singlet excited state with an emission that is centered around 1270 nm (~ 0.97 eV). Please note, however, that this does not necessarily place the triplet excited state energy of **Tr*[10]** at a value of 0.97 eV. Instead, it places the triplet excited state energy *above* 0.97 eV to render the underlying energy transfer from **Tr*[10]** to the triplet ground state of oxygen thermodynamically feasible.

Reviewer comment:

Why is not the fluorescence scanned to higher energies for **Tr*[6]** and **Tr*[8]** (as it is done with **Tr*[10]**)? I think information is missing because of this and cause the information on the Stokes shift in **Table S3** to be erroneous. Furthermore, please state excitation wavelength and detection wavelength in the caption.

Response to Reviewer comment:

We would like to thank the reviewer for this valuable comment. Actually, the fluorescence of **Tr*[6]** and **Tr*[8]** have also been recorded to 310 and 340 nm, respectively – see the provided review-only Figure below.

Tr*[6]:
Excitation: 290 nm
Raman band relative to excitation: 2887 cm^{-1}

Tr*[8]:
Excitation at 320 nm
Raman band relative to excitation: 2921 cm^{-1}

Tr*[10]:
Excitation at 330 nm
Raman band relative to excitation: 2906 cm^{-1}

Nevertheless, the overall low fluorescence intensity of any of the oligynes and the presence of hexane-related Raman features, which are $2800\text{--}3000\text{ cm}^{-1}$ red-shifted relative to the excitation wavelength, hampered the unambiguous assignment of the fluorescence peaks in the high-energy region for **Tr*[6]** and **Tr*[8]**.

Regarding **Table S2** (which we assume the reviewer was referring to), we were indeed unable to exclude additional fluorescence-related peaks for **Tr*[6]** and **Tr*[8]** in the high energy region, but they are very likely masked Raman-related features. In light of these considerations, we have replaced the values for Stokes shifts in **Table S2** by “n.d.” and have added a

corresponding footnote to the revised version of the manuscript. The Figure shown above was inserted into the *Supplementary Information* as **Figure S3**. Furthermore, the corresponding excitation and emission wavelengths have been added to the caption of **Figure 2**.

Reviewer comment:

Page 7: “reflecting the larger degree of conjugation in longer oligoynes”, can the authors explain why. The IR signal can go to lower energies either because an increase in effective mass of the vibrating unit, or a decrease in binding energy. As I see it, if something is reflecting a higher degree of conjugation, then two different vibrations (the carbon-carbon single bond and the triple bond) would gradually merge to one single vibration.

Response to Reviewer comment:

We acknowledge the Reviewer’s criticism and have revised the text on page 7 accordingly. Moreover, we direct the reader to the appropriate studies that have investigated the relationships between conjugation length and vibrational absorptions in oligoynes (new reference 49, Agarwal *et al. J. Raman Spectrosc.* **44**, 1398–1410 (2013)).

Reviewer comment:

Page 8: “from the longest wavelength absorption and shortest wavelength fluorescence...” The authors show that the forbidden $S_0 \rightarrow S_{2/3}$ state do absorb light, therefore is this statement erroneous.

Response to Reviewer comment:

We agree with the Reviewer and apologize for this erroneous statement. We have corrected the statement to “*from the $S_0 \rightarrow S_n$ absorptions and shortest wavelength fluorescence ...*”.

Reviewer comment:

Why were the two series excited at different wavelength in the fs TA experiments? Was there a rational for the excitation wavelength? Furthermore, can the forbidden $S_0 \rightarrow S_1$ be excited directly? A lack of the initial decay time when analyzing a decay after direct excitation would significantly strengthen the mechanism suggested in the TA measurements.

Response to Reviewer comment

As described in the *Supplementary Information* (3. General Procedures and Methods), the transient absorption data were collected in two different labs for the **Tr*[n]** and the **Glu[n]** series, respectively. On one hand, the **Glu[n]** series was probed with an excitation wavelength of 390 nm, which was generated by frequency doubling of the 780-nm fundamental of the laser, and, on the other hand, the **Tr*[n]** series was excited with 258-nm pulses, which were generated by frequency tripling of the 775-nm fundamental of the used laser system. In short, the only rational for the different excitation wavelengths was the varying laser configurations in the two different labs. To corroborate the wavelength-independent character of the suggested mechanism, **Tr*[10]** was also excited at 387 nm. As shown in the Figure below, which we also implemented as **Figure S12** in the *Supplementary Information*, the resulting evolution-associated fits of the spectra yield very similar species for each state. We are therefore confident that none of the other **Tr*[n]** and **Glu[n]** are subject to any dependence on the excitation wavelength. We appreciate the suggestion to directly excite the forbidden $S_0 \rightarrow S_1$ transition to prevent the initial transformation of $S_n/S_{2/3}$ into S_1 and to observe this via the lack of this component in the corresponding transient absorption spectra. However, we

seriously doubt to that a direct $S_0 \rightarrow S_1$ excitation while excluding the simultaneous excitation into higher states ($S_n/S_{2/3}$) is feasible.

REVIEWERS' COMMENTS:

Reviewer #1 (Remarks to the Author):

I have carefully examined the replies and corresponding changes regarding my comments suggestions. As a result, most of them are satisfactory. However, though I have seen the authors' reply to my comment "I do not see relevant statements to elaborate if these findings could constitute a significant advance in future progress either in fundamental or applications.", I do not see anywhere that the authors address my comment (as their replies in the letter) in the revised text. If this can be done then I recommend its publication.

Reviewer #3 (Remarks to the Author):

In my opinion, the technical quality is high enough for publication.

Submission of Revised Manuscript NCOMMS-20-03672-T – Response to Referee Comments

Response to Reviewer 1:

Reviewer comment:

I have carefully examined the replies and corresponding changes regarding my comments suggestions. As a result, most of them are satisfactory.

Response to Reviewer comment:

We would like to thank the reviewer for his/her very positive assessment of the revised manuscript. We sincerely hope that the answer provided below meets the remaining concern.

Reviewer comments:

However, though I have seen the authors' reply to my comment "I do not see relevant statements to elaborate if these findings could constitute a significant advance in future progress either in fundamental or applications.", I do not see anywhere that the authors address my comment (as their replies in the letter) in the revised text. If this can be done then I recommend its publication.

Response to Reviewer comment:

We have added the following statement regarding the relevance of our findings to the conclusions of our manuscript:

The results of the present study provide more accurate and experimentally verified predictions for the photophysical properties of molecules with extended π -conjugated systems based on sp-hybridized carbons that will help guide the preparation of carbon-rich materials with tailored properties in the future.

We sincerely hope that this highlights the relevance of our findings in light of fundamental aspects as well as the potential broader implications for future applications.

Response to Reviewer 3:

Reviewer comment:

In my opinion, the technical quality is high enough for publication.

Response to Reviewer comment:

We would like to thank the reviewer for his/her positive assessment.